# Modeling Drivers' Situational Awareness
# from Eye Gaze for Driving Assistance

**Abhijat Biswas**    **Pranay Gupta**    **Shreeya Khurana**

**David Held**    **Henny Admoni**

Robotics Institute
Carnegie Mellon University, United States
Correspondence: abhijatbiswas@gmail.com

**Abstract:**

Intelligent driving assistance can alert drivers to objects in their environment; however, such systems require a model of drivers' situational awareness (SA) (what aspects of the scene they are already aware of) to avoid unnecessary alerts. Moreover, collecting the data to train such an SA model is challenging: being an internal human cognitive state, driver SA is difficult to measure, and non-verbal signals such as eye gaze are some of the only outward manifestations of it. Traditional methods to obtain SA labels rely on probes that result in sparse, intermittent SA labels unsuitable for modeling a dense, temporally correlated process via machine learning. We propose a novel interactive labeling protocol that captures dense, continuous SA labels and use it to collect an object-level SA dataset in a VR driving simulator. Our dataset comprises 20 unique drivers' SA labels, driving data, and gaze (over 320 minutes of driving) which will be made public. Additionally, we train an SA model from this data, formulating the object-level driver SA prediction problem as a semantic segmentation problem. Our formulation allows all objects in a scene at a timestep to be processed simultaneously, leveraging global scene context and local gaze-object relationships together. Our experiments show that this formulation leads to improved performance over common sense baselines and prior art on the SA prediction task.

**Keywords:** situational awareness, eye gaze, driving assistance

## 1  Introduction

Future Advanced Driving Assistance Systems (ADAS) might include driver assistance systems that warn users about objects in their environment that they should pay attention to. Imagine a system that runs on your intelligent vehicle while you drive, tracking important traffic objects like vehicles and pedestrians [1]. Such a system could conceivably warn you about objects that are likely to be in your path or are otherwise dangerous, improving safety for everyone on the road. However, you are not very likely to adopt such a system if it alerts you about every object on the road regardless of your awareness of it — a well documented phenomenon known as "alert fatigue" [2]. To address this gap, we tackle the real-time object-level modeling of drivers' Situational Awareness (SA) [3], specifically the set of traffic objects (vehicles, pedestrians, and two-wheelers) in the world that the driver is aware of at any given time.

Drivers' eye gaze is closely linked to their situational awareness [4, 5, 6]. However, inferring situational awareness from eye gaze is not as simple as just counting gazed-at objects, since we regularly use our peripheral vision and memory to build and maintain situational awareness while driving [4].

---

Our code and dataset are available at https://HARPLab.github.io//DriverSA

8th Conference on Robot Learning (CoRL 2024), Munich, Germany.

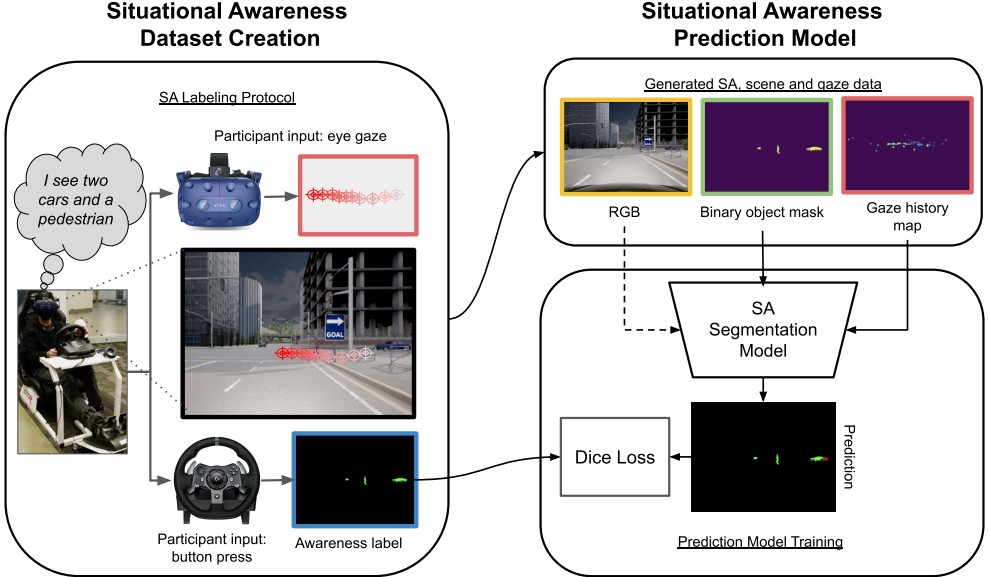

Figure 1: We collect drivers' object-level situational awareness (SA) data via a novel interactive protocol in a VR driving simulator. We use the generated data to train a driver SA predictor from visual scene context and driver eye gaze. Casting this as a semantic segmentation problem allows our model to use global scene context and local gaze-object relationships together, processing the whole scene at once regardless of the number of objects present.

Additionally, drivers can ostensibly "gaze" at objects without gaining situational awareness of them, due to effects like inattentional blindness or saccading over objects without fixating on them [7].

Thus, we aim to learn a supervised model for predicting a driver's situational awareness from their eye gaze and the scene context. However, training such a model requires a driving dataset with explicitly labeled drivers' object-level situation awareness. This dataset should be a collection of sequences of driving events comprising the scene context, the driver eye gaze history over the scene, and labels of the drivers' situational awareness over each traffic object.

To be useful for machine learning and the downstream assistance tasks, there are a few key desiderata for these awareness labels: 1. Labels should explicitly denote the start of the drivers' awareness over each object and hence be continuous. This is important since the transition of driver awareness is crucial for determining when it is appropriate to alert the driver to the object. 2. Labels should be dense over the set of traffic objects, i.e. we want a label for every traffic object that enters the driver's field of view. 3. Labels should be collected in a way that does not affect the normal gaze behavior of the driver to avoid distribution shift between training and deployment gaze behavior.

Obtaining object-level awareness labels with *all* the aforementioned properties simultaneously is challenging for a few reasons. Most current SA labeling efforts collect data either intermittently or sparsely [8, 9, 10, 11, 12]. For instance, the common Situation Awareness Global Assessment Technique (SAGAT) [13, 6] involves freezing and blanking the screen during occasional pauses in simulated driving, followed by probing the driver about traffic objects present in the scene. These collected labels are intermittent — they are valid for the moment when the simulation was paused, but do not tell us when a driver first becomes aware of an object. Furthermore, these labels are sparse, as the driver is only probed about objects within certain parts of the scene.

In this work, we introduce a novel SA labeling protocol (Sec. 3) that produces continuous and dense object-level SA labels. As a part of our protocol, drivers indicate their awareness of all objects in their field-of-view, by pressing directional buttons on the steering wheel controller (Fig. 1). We collect a dataset of 80 episodes using our protocol. In each episode, drivers are instructed to drive to an in-world goal inside a VR driving simulator [14] while following the SA labeling protocol. We record their driving actions, eye gaze, and SA labeling button presses along with the simulator state.

Further, we use the aforementioned dataset to learn a model that predicts a drivers' object-level SA status given the scene context and a history of the driver's eye gaze (Sec. 4). We cast this problem as a semantic segmentation problem and show that it performs better than a common-sense gaze-intersection baseline and prior work that uses handcrafted features [6]. Our formulation allows us to process a variable number of objects in the scene in a single inference step as opposed to prior work which processes each object in a scene separately, repeating global computations.

In summary, our contributions (Fig. 1) are the following:

- **SA Labeling Protocol**: an interactive protocol for obtaining continuous and dense SA labels for on-road agents in a driving scene, without disrupting the driving task
- **SA Data Collection**: a driving dataset with continuous object-level SA labels, traffic object states, and driver eye gaze collected using our protocol in a VR driving simulator with 20 drivers
- **SA Prediction Model**: a learned gaze-based driver situational awareness model which predicts SA over the scene on an object-level basis

Our code and dataset will be released publicly upon acceptance.

## 2 Related Work

**Measuring Situational Awareness:** Determining a driver's internal awareness of the environment and traffic objects (vehicles, two-wheelers and pedestrians) is challenging due to our use of peripheral vision and behaviors like intentional blindness or saccading [15]. Prior approaches for extracting information about a driver's internal awareness involve collecting data intermittently or sparsely. An example of this is the Situation Awareness Global Assessment Technique (SAGAT), used by prior work to collect dense object-level SA labels from drivers [6]. This involved periodically pausing the simulated driving scenario, blanking the screen, and then asking the driver a series of questions about their awareness of individual objects in the scene. Another approach, called Daze [16], mitigates some SAGAT issues by posing real-time queries about recent events without pausing the simulation. However, it does not yield dense object-level labels and requires looking away from the driving scene to answer affecting natural eye-gaze behavior. An influential indirect technique is the Situation Present Assessment Method (SPAM) [9], which uses real-time verbal probes about past, present, and future situations to indirectly measure SA based on response accuracy and latency. SPAM importantly also uses response times as an index of how readily this information is available. For our requirements, verbal queries have the same label sparsity issue as Daze as well as requiring manual post-processing to get machine readable annotations from verbal responses.

**Driver Situational Awareness Models:** Using eye gaze to infer driver attention and awareness are not new ideas, with preliminary studies having been around since at least the 1906s [17]. However, using these signals together with outward scene context for driver assistance is a relatively new area enabled by advances in sensor quality, form factors, and onboard computation —with the first papers appearing in the late-2000s [18]. Initial work used signals such as gaze direction in discrete traffic-facing zones as a crude proxy for driver attention to determine if traffic objects were causing distracted gaze. More recently, the paradigm has been to match driver gaze to objects in the traffic scene to determine whether the driver has noticed them and raise an alert when necessary [15].

We will focus our discussion on the process of matching gaze to traffic objects to determine which ones the driver is aware of. A naive solution is to simply count objects whose bounding boxes contain driver gaze points [19]. However objects can be perceived without being directly gazed at and 3D gaze direction estimation can have errors [20]. More recently, hand-designed feature based learning methods have emerged [13] that predict the driver's attention given a history of their gaze relative to traffic objects. Some such methods even account for concepts of working memory from psychology [6]. However, evaluating these methods against one another is challenging. Some of these methods were evaluated qualitatively without any objective ground truth being present (SA ground truth is hard to collect as discussed in the previous section) [21]. Other methods have only been evaluated offline and on data collected using SAGAT, meaning they are evaluated on singular snapshots rather than a stream of driving data [13, 6] which prevents important aspects like aware-

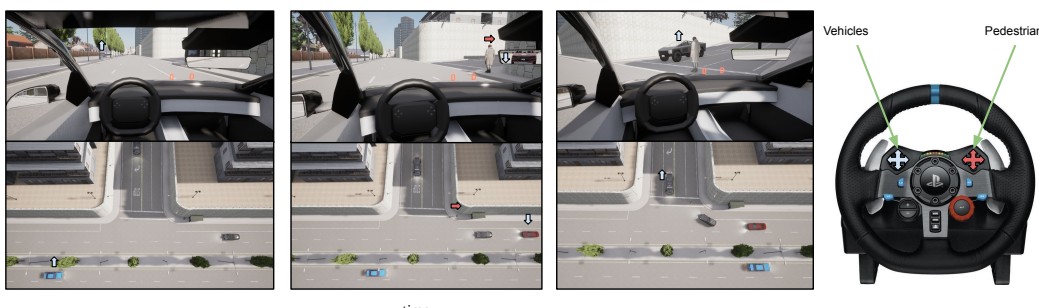

Figure 2: Example sequence of right hand turn with object-level driver responses. The top row shows the scene from the driver view and the bottom row shows the same scene via a birds-eye view. Labels are shown as colored arrows above the respective traffic object. Labels correspond to buttons on the steering wheel (right). Blue corresponds to vehicle labels and red to pedestrian labels.

ness transition points to be represented in the data. Their data and models are also not publicly available, making comparative evaluation difficult. To help mitigate this issue for future research, we will release our continuously-labeled SA dataset publicly.

## 3   Situational Awareness Data Collection

We collected our driver object-level SA dataset in a VR driving simulator (DReyeVR [14]). Drivers were asked to drive safely following a series of directional goal signs (see RGB image in Fig. 1) along scripted routes. The drives were instructed to simultaneously follow the SA labeling protocol to record object-level SA labels.

**Situational Awareness Labeling Protocol:** Under our proposed SA labeling protocol, drivers are instructed to push a button on their steering wheel as soon as they perceive a vehicle, pedestrian, or two-wheeler (collectively, traffic objects). For each new traffic object they perceive, they are instructed to press one of four buttons to indicate their awareness (see Fig. 2). The button to be pressed is determined by the relative position of the target object to the ego-vehicle. For instance, if there is an object in front of the vehicle, the forward button should be pressed. The steering wheel used has two sets of four buttons; the set of buttons on the left is used for vehicles and the right one is used for 2-wheelers+pedestrians. An example sequence of traffic objects and their corresponding button presses is shown in Fig. 2.

The awareness labels are generated by associating button clicks with target objects. The direction is used to associate button presses with target objects. An object in a scene is considered 'unaware' until it is associated with a button press, after which it's status is flipped to 'aware'. More details about how the awareness labels are generated can be found in the supplementary material.

**Route & traffic design:** Each route consists of a predefined source, destination, and path. Each route also contains in-world navigational goal signs to direct the drivers along the path. Routes were designed to have an average drive time of about 4 minutes. Each route was driven by a maximum of 8 drivers and a minimum of 4 drivers and there were a total of 15 unique routes. Participants were pre-assigned routes so each route would be seen equally but some chose to terminate early due to VR-induced nausea, causing an imbalance in the final number of routes.

At least one safety critical scenario such as a jaywalking pedestrian was included in each route. We did so to ensure that driver gaze before and during safety critical scenarios was also represented in the dataset. More details on the scenarios can be found in the supplementary material. The traffic along each route was randomly generated. However, multiple objects appearing in the scene from any single direction could lead to ambiguities in associating objects with button clicks. Hence, we limit the number of new objects of each type appearing simultaneously at intersections in each direction to one. Note that having different sets of buttons for vehicles and pedestrians(+two wheelers) allows us to disambiguate between object types appearing in the same direction.

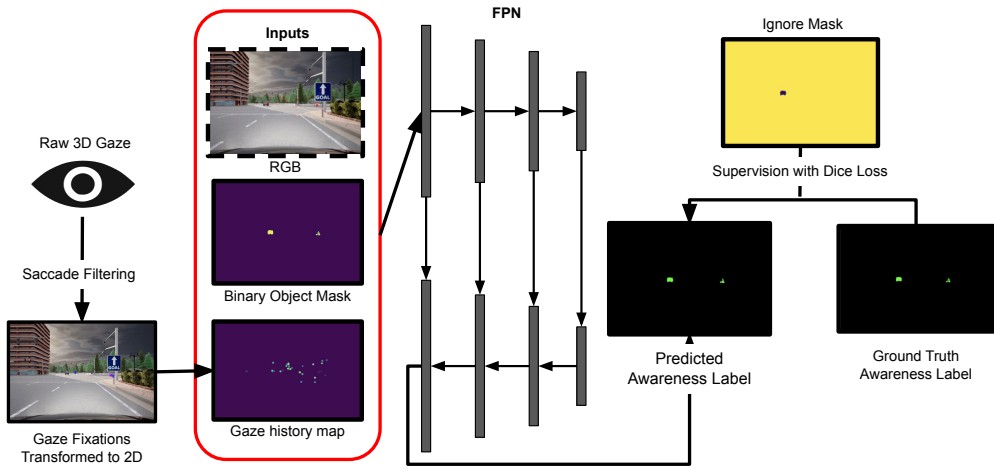

Figure 3: Object-wise SA prediction algorithm. A history of raw driver gaze is filtered to exclude saccades and then transformed to 2D pixels in the current camera position. These are used to create a gaze history map which is input together with an object segmentation of the scene (or optionally, RGB). The Feature Pyramid Network (FPN) then produces a 3 class segmentation (unaware, aware, background). During training, loss is ignored for objects which entered into the driver awareness prior to the gaze history window.

**Data collection details:** We ran our SA protocol with 20 participants, each with 1+ year of holding a valid US or international driver's license. Each participant was given a set of scripted instructions and were first given time to interact and familiarize themselves with the interface and the simulator. Once they were comfortable with driving in the simulator, they were introduced to the secondary labeling task and asked to perform it while completing a trial route. Participants saw a maximum 5 non-trial routes each, but some participants did not complete all 5 routes due to the onset of discomfort from VR cybersickness. We collected a total of 80 routes worth of data which resulted in about 340 minutes of recorded driving time. The data collection was approved by the university's IRB. Some additional details about the data collection are provided in the supplementary material.

## 4 Modeling Driver SA

In modeling driver situational awareness, our goal is to predict a driver's awareness status over all dynamic traffic objects in the scene at a given time using scene information in conjuction with the driver's gaze. Specifically, for any given traffic object $obj$, we would like to produce a prediction of the binary awareness status $A_{obj}$ where $A_{obj} \in \{aware, unaware\}$.

**Problem formulation:** We cast the problem of driver SA modeling as a segmentation problem, where the input is a visual representation of the scene in front of the user and the user's gaze, and the output is a prediction of the objects in the scene that the driver is aware of.

The scene is represented by a binary object mask indicating the location of objects in the scene (see "Visual scene representation" below for details); the user's eye-gaze history is input as an additional channel in the same spatial coordinates (see "Gaze history map" in Figure 3). Under our formulation, each timestep $t$ represents a data point where the observations are an object mask of the scene and a gaze map: $O_t = (I_t^{obj} \in \mathbb{R}^{600 \times 800}, I_t^{gaze} \in \mathbb{R}^{600 \times 800})$. The output of our model is a segmentation map with 3 classes: *aware, unaware, & background*. Object-level awareness labels are then derived from the output segmentation by taking the mode class of the pixels corresponding to each object while ignoring the background class, giving us $A_{obj}$ for each object that is visible in $O_t$.

Alternative formulations could see this posed as a classification problem, where each object is a data point and the neural network is trained to predict a single object-level awareness label instead. In contrast, our formulation requires one forward pass per timestep, rather than once per target object in a timestep. This avoids repeated computations since the objects share their global context.

**Gaze representation:** The gaze history map $I_t^{gaze}$ is obtained from a sequence of 3D gaze over a historical window of length $W$ seconds. If we sample gaze at a rate of $s$ Hz over this window, we obtain $N_g = s \times W$ samples of gaze. Each gaze sample is a 3D ray $G_i$ pointing in the direction of the driver's gaze, which we project into the camera coordinates to convert to a 2D pixel location. The 3D point on this ray we project into 2D is the first point of intersection with the world while ignoring the ego vehicle mesh (since the ego-vehicle windshield is not the point of interest). We transform the gaze into 2D pixel coordinates $g_i = M_t G_i \ \forall \ i \in \{1, 2, ..., N_g\}$, where $M_t$ represents a transform from world coordinates to the coordinates of the camera used at timestep $t$. Note that this transformation accounts for the current pose of the ego-vehicle at time $t$ such that the historical 3D gaze points are transformed into pixels corresponding to their location at that previous timestep. This means that sometimes older gaze points are out of the frame due to the traffic object's subsequent motion. In our experiments, we use a gaze window of $W = 10$ s.

Gaze pixel locations are represented as a fixed size dot (see "Gaze history map" in Figure 3). We also perform an ablation with a heatmap-based representation as is common with other literature (e.g. [22]) but found this to perform worse (see Sec. 5). To include a sense of temporality in the gaze, we fade the value of the gaze dot linearly from 255 to 10 across the window so that the most recent gaze dots are the brightest. Additionally, since drivers cannot gain new awareness during gaze saccades (see saccadic suppression, Ch 2. [23]), we perform gaze event detection using the I-BMM classifier [24] and exclude saccades from the gaze map.

We also use an additional "ignore mask" to zero out losses from traffic objects that entered the user's awareness more than $W$ seconds ago. Consider a vehicle that entered the user's awareness 15 s prior to the current timestep. If we use a history window $W = 10$ s, the driver gaze correlated with awareness of that vehicle is no longer represented, though the vehicle is still labeled as *aware*. If we penalize the network during training for mis-classifying that object, we are penalizing a prediction for which the network has incomplete information.

**Visual scene representation:** The visual scene representation uses a binary object mask to represent the scene; the mask indicates the location of relevant dynamic traffic objects: vehicles, pedestrians, and two-wheelers. We choose to use a fixed size ($600 \times 800$) image representation from a viewpoint in front of the ego-vehicle to control the scope of experiments. However, due to our formulation as a segmentation problem, our model can deal with arbitrarily sized inputs. This can be useful, for instance, when using wider aspect ratio visual inputs to represent the wide field of view that human drivers naturally have. The binary object mask was obtained directly from CARLA, but could be replaced by any off-the-shelf vehicle/pedestrian segmentation algorithm.

**Model and training details:** We used a Feature Pyramid Network [25] segmentation model with a MobileNetV2 [26] backbone (pre-trained on ImageNet). The backbone was chosen for its low number of parameters ($2M$) and runtime efficiency. While our dataset contained a similar number of aware to unaware objects, unaware objects usually were further from the ego-vehicle and occupied much smaller portions of the input images. We calculated the ratio of the unaware pixels to aware pixels in the label masks as about 1:20 and used an unaware class weight of 20 (background weight=$10^{-5}$). We trained with the Dice loss due to its ability to handle class imbalanced data [27].

## 5 Evaluation & Discussion

**Baselines:** We compare our method to three baselines: the majority class, a common-sense gaze intersection baseline, and a prior art baseline using handcrafted features. The "**majority class**" baseline simply predicts the majority class in the test set ("unaware": $53\%$ share). The "**gaze intersection**" baseline performs a simple check: if the driver's gaze is within the segmentation mask of a traffic object (vehicle, pedestrian, or 2-wheeler) in the past $T$ seconds, it assigns the *aware* label to it (others assigned *unaware*). We use $T = 10$, matching the other baselines.

The prior art baseline ("**handcrafted features**") is an SVM model that takes several handcrafted features as input and produces a binary label output [6]. We re-implemented their model based on

| Model | inf. cmplx. | Acc. ($\uparrow$) | Prec. ($\uparrow$) | Recall ($\uparrow$) |
|---|---|---|---|---|
| Majority class | 1 | 52.99% | 0.53 | **1** |
| Gaze intersection | 1 | 46.87% | 0.41 | 0.54 |
| Handcrafted features [6] | N | 65.47% | 0.66 | 0.69 |
| Ours | 1 | **72.33%** | **0.73** | 0.77 |

(a) Performance of our model & baselines

| Model Ablation | Acc. | Prec. | Recall |
|---|---|---|---|
| No ignore mask | 69.04% | **0.80** | 0.58 |
| Raw gaze | 66.67% | 0.76 | 0.57 |
| Gaze heatmap | 69.30% | 0.77 | 0.62 |
| No gaze fading | 69.73% | 0.76 | 0.66 |
| Gaze $20s$ hist. | 69.27% | 0.78 | 0.57 |
| Gaze $5s$ hist. | 64.79% | 0.68 | 0.68 |
| RGB | 54.31% | 0.71 | 0.29 |
| Ours (Full) | **72.21%** | 0.73 | **0.77** |

(b) Ablations for our model

Table 1: Prediction performance of models and baselines on the SA prediction task. Our model outperforms the non-trivial baselines on all 3 metrics and scales better as objects in the scene increase. [inf. cpmlx. = inference time complexity with N objects, Acc. = Accuracy, Prec. = Precision]

the paper description (authors' code or data were not publicly available). The original work lists 5 sets of features, computed across a $10s$ analysis window (similar to the gaze history window in our method): *Gaze point-based*, *Human visual sensory dependent*, *Object spatial-based*, *Object property-based*, and *Human short-term memory-based*. We implemented the first 3 of these feature sets and the object type feature (vehicle vs pedestrian) from the "Object property-based" set. Most of the "Object property-based" features were excluded since they were difficult to compute and required privileged scene information (*e.g.* one feature required the state of the corresponding traffic light for every traffic object in scene; another was manually annotated). Human short-term memory-based features were also excluded since they were difficult to compute and did not contribute much ($< 1\%$ point) to overall performance in the original evaluation [6]. The original SVM was trained on $1078$ training samples. Since neither the trained model nor code were available, we trained our implementation of the SVM on a subset of our training data. We trained the SVM on 10 episodes in our train set, which is about $3\times$ the training data used in the original work. SVM implementations generally cannot handle very large datasets since the entire dataset is loaded into memory during training and mini-batch SVM training is non-trivial. To train the SVM, we used a machine with 128GB RAM but could only use $15\%$ of the training set.

**Experimental settings:** Our dataset contains 80 episodes of which we used 64 ($80\%$) for training. $10\%$ of the training episodes were used as the validation set. The test set was a separately held out set of 16 episodes. It was partitioned so that participants were disjoint between the train and test set. This is important since we want to test the generalization to new users; it would be impractical to put every new driver through the SA protocol when deploying such a system.

We use 3 metrics to evaluate and compare methods: object-level accuracy, precision, and recall. For precision and recall, the positive class is the "unaware" class. This is because downstream applications such as driver assistance systems which alert the driver will care about how well our system can predict which traffic objects the driver is not aware of. "Precision" is thus a measure of how often our prediction of an object being unaware is correct — errors are "aware" objects classified as "unaware." This type of error can lead to alert fatigue for an end-user [2]. "Recall," on the other hand, indicates how many of the "unaware" objects in the dataset were correctly predicted — these are objects that the driver wasn't aware of but our system predicted that they were.

**Results & Discussion** Our quantitative evaluation results can be found in Table 1. The naive gaze-intersection baseline, as expected, performs the worst, confirming that it is not enough to simply count which objects were "gazed-at". The prior art handcrafted features baseline performs better but significantly worse than our method. In terms of runtime, the prior art baseline has 2 expensive parts: computing features per object and doing SVM inference (this can be batched across objects). On average each part takes 5 ms, resulting in a total average runtime of $(5N + 5)$ms on an AMD 5955WX CPU (for N objects in scene). In contrast, our network takes 11ms total for a forward pass (on a 4090 GPU) and does not scale with the number of scene objects. Some qualitative comparisons of model outputs can be seen in Fig. 4.

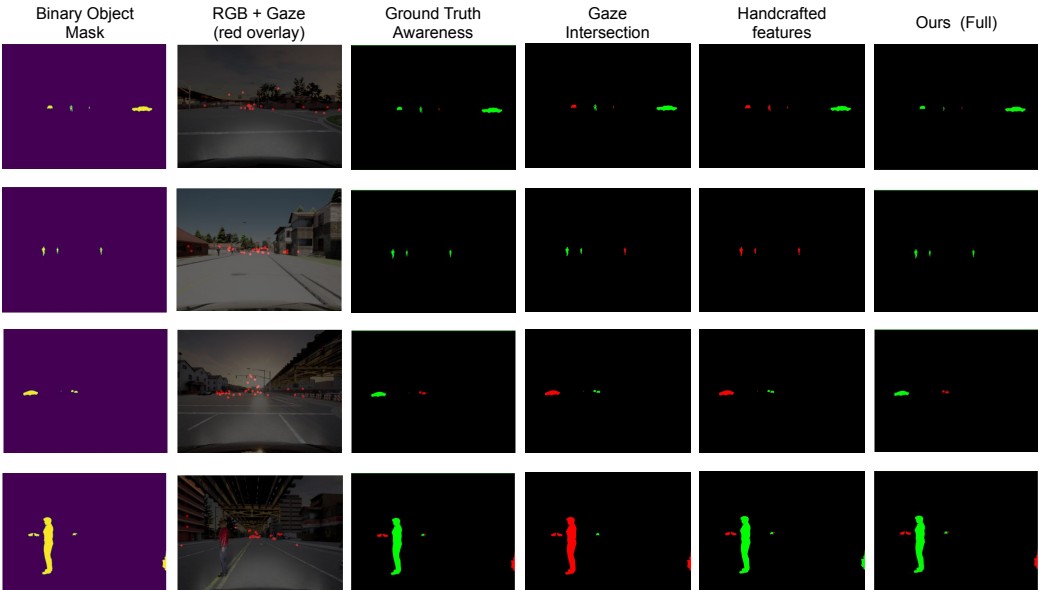

Figure 4: Qualitative results for our model and baselines. Each row represents an independent driving scene. The RGB image shows the most recent $10s$ of gaze overlaid as red dots.

Our ablations (Table 1, right) show the performance impact of several design choices described in Sec. 4. In terms of gaze representation, the ignore mask (used to avoid penalizing mispredictions of awareness transitions outside the gaze history window) was the most important during training — responsible for an $8\%$ accuracy drop when removed. Using saccade filtered gaze instead of raw gaze was the next most important. We also investigated the use of gaze heatmaps as the gaze representation similar to previous work [22, 28], in which each gaze point is represented by an isometric 2D Gaussian that could accumulate in weight at fixations; this performed about $3\%$ worse than using fixed sized dots. This is similar to the issue of representing corrective clicks in an interactive segmentation task, where a similar result has been found [29]. The results indicate that the use of gaze fading was only responsible for about $2\%$ of the model's performance. This suggests that the presence and location of a gaze point within the gaze history window contains most of the information about awareness rather than the exact temporal order of the gaze. Finally, using an RGB image as input resulted in $20\%$ worse accuracy with the same model size (except the initial layer), as the model now has to simultaneously perform segmentation and SA modeling.

**Limitations:** Our proposed SA labeling protocol is mainly limited by the fact that some traffic configurations can lead to ambiguity in assigning a button — whenever there is more than one new object of the same type (vehicle or pedestrian) from the same cardinal direction relative to the driver. We created an interface for manual annotation to resolve ambiguities post-hoc. The biggest limitation of our model is its static, memoryless nature. Since SA is inherently a temporal signal, improvements can probably be achieved by performing temporal modeling. Currently, our method treats each timestep as independent and would require an external module to implement memory.

## 6    Conclusion & Future Work

We proposed a new interactive protocol to record human drivers' object-level situational awareness that produces continuous and dense awareness labels. We use it to record a SA dataset with 20 drivers in a VR driving simulator. Additionally, we use this dataset to train a driver object-level SA model by casting it as a semantic segmentation problem. Our model outperforms baselines and prior work while scaling better to arbitrary numbers of objects in the scene. In the future, we plan to use our driver SA model in the inner loop of a driver assistance system that provides intelligent alerts or interventions in safety-critical situations and evaluate this in a simulator-based user study. We also commit to releasing our code and data publicly upon acceptance in the hope that it will facilitate more work in the domain.

**Acknowledgments**

This work was partially supported by the Link and Tang foundations, and the National Science Foundation (IIS-1943072).

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

# A Additional Related Work

## A.1 Situational Awareness: definitions from aviation to driving

First popularized by Mica Endsley's work in aviation, pilots' SA was defined as "the perception of the elements in the environment within a volume of time and space, the comprehension of their meaning, and the projection of their status in the near future" [8]. According to Endsley, SA reflects the extent to which the operator knows what is going on in their environment and is the product of mental processes including attention, perception, memory, and expectation [30]. This definition laid out three levels of SA: (1) perception (of situational elements) , (2) comprehension (of their semantics), and (3) projection (of their futures states). In the original aviation context, these elements comprised instruments and instrument panels that pilots needed to maintain SA over in order to perform the aviation task safely and successfully. However, in the driving context these scene elements not only comprise similar in-vehicle instruments such as the speedometer and rear-view mirrors, but also outside-the-vehicle elements such as other vehicles, bicycles, pedestrians etc. For tracking with respect to pilot/driver eye gaze, a functionally challenging difference among these elements is that the driving elements constantly change position relative to the vehicle while the aviation instruments are fixed and their locations are known. This difference makes is difficult to apply techniques (for grounding, evaluation etc.) from aviation directly to the driving case.

## A.2 Situational Awareness labeling methods

At a high level, situation awareness (SA) grounding methods can be classified into direct (e.g. queries about objects for which SA is estimated) and indirect (SA inferred from secondary task measures such as response time to probes). As we discuss these, we will comment on the suitability of these techniques to generate per-object labels for learning a gaze-based per-object SA model.

| SA Labeling Method | Capture Awareness Transition | Dense Object Labels | Doesn't Natural Behaviour | Affect Gaze |
|---|---|---|---|---|
| SAGAT [8] | ✗ | ✓ | ✓ | |
| DAZE [16] | ✓ | ✗ | ✗ | |
| SPAM [9] | ✓ | ✗ | ✗ | |
| Ours | ✓ | ✓ | ✓ | |

Table 2: Our SA labeling protocol allows us to capture the transition in the driver's awareness of objects in the scene, allows labels for all objects in the scene without affecting the natural gaze behaviour of the driver.

### A.2.1 Direct methods

Within direct methods, we may classify grounding techniques into objective or subjective based on whether the probes involve questions about directly measureable quantities (e.g. number of red vehicles around you) or self-rated ones (e.g. perceived task load). We will first discuss objective measures. Perhaps the most well known and used direct objective method of Situational Awareness grounding is the Situation Awareness Global Assessment Technique (SAGAT) [31]. The SAGAT involves operators performing a simulated version of a real task such as driving. Intermittently, the simulation is paused (the screen can be blanked or only the background is presented) and the operators are asked several questions about the situation right before the pause. Accuracy of responses to these questions determines the operators' SA. SAGAT was first designed for aviation but has been adapted to driving [6]. Despite its popularity, SAGAT has its limitations mainly associated with the mandatory simulation pauses required. There are cognitive process modifications to the normal task because of removal from the task during the probe as well as intermittent task resumption deviations [12]. For generating ground truth data for per-object SA, we also have some issues. One,

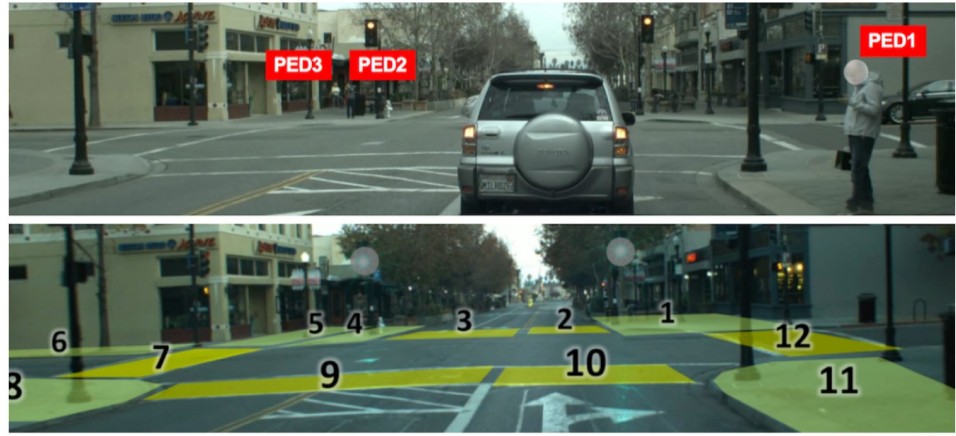

(a) SAGAT freezes simulations or videos being watched (top) and then asks participants the location of traffic elements (bottom). Image from [6].

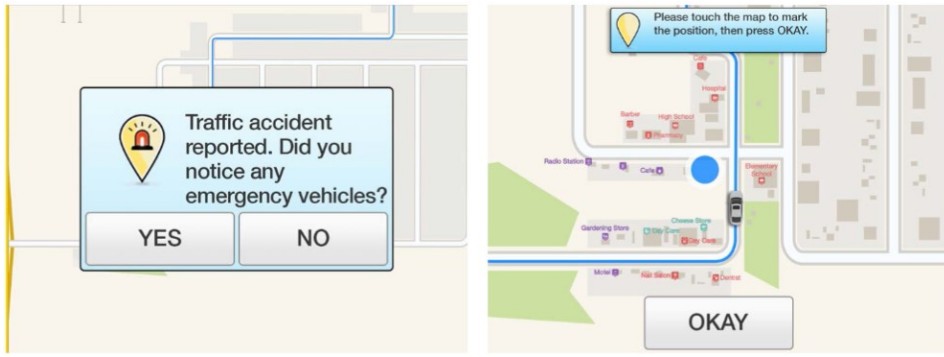

(b) DAZE does not require pauses. It asks participants if they noticed particular types of traffic elements and to mark their locations on an overhead GPS map. Image from [16].

Figure 5: Examples of SA labeling methods used in previous work. These methods produce intermittent labels (SAGAT/DAZE) or sparse ones (DAZE —not every object is labeled).

we only get SA labels per queried object at the time of the probe —SAGAT probes do not give us the starting point of the operators' SA for each queried object. Second, SAGAT querying requires pauses hence limiting the number of labels per drive that could be collected while maintaining the flow of simulation.

Another direct objective measure that mitigates some of these issues is Daze [16] which uses real-time in situ questions that resemble queries drivers are already familiar with (such as traffic queries from apps like Waze). In particular, shortly after an on-road event such as an accident has passed, it raises an alert asking a question such as "Traffic accident reported. Did you notice any emergency vehicles?". While this method avoids pausing the simulation (an indeed can also be used for on-road driving), it does not provide dense, per-object labels in the way we require. Additionally, answering the query involves looking away from the driving scene and at a tablet or screen which undesirably modifies gaze behavior.

In conjunction with objective methods, subjective measurements can be useful. For example, operators' perceived estimate of their own SA may important in determining their actions or interactions with an SA enhancing system. Here, we will only discuss the most commonly used subjective measure: Situational Awareness Rating Technique (SART). SART is administered as a 14-part post-hoc questionnaire in which, operators rate on a series of bipolar scales the degree to which they perceive (1) a demand on their resources, (2) supply of operator resources and (3) understanding of the situation. These are combined to provide an overall SART score [11]. However, there are limitations to SART as a measure of the operators' SA. For example, consider unknowingly unknown scene ele-

ments: operators cannot rate their SA on all scene elements if they didn't know they missed some. Other factors are the influence of performance on SART, as well as confounding with workload [32].

### A.2.2 Indirect methods

Within indirect SA grounding techniques, the most widely accepted protocol is the Situation Present Awareness Method (SPAM) [9]. SPAM involves a real-time probe (usually a verbal query about the past, present, and future aspects of the situation) while the operator is performing their primary task. While direct measures such as response accuracy are collected, SPAM importantly also uses response times as an index of how readily this information is available. For our requirements, verbal queries have the same label sparsity issue as Daze as well as requiring manual post-processing to get machine readable annotations from verbal responses.

### A.2.3 Physiological methods

For the sake of completeness we must mention the use of physiological methods in the literature to measure operator SA. These signals have the benefit of being continuous variables rather than isolated or posthoc probes mentioned above. These methods have employed physiological signals such as EEG [33], respiratory rate [34], and heart rate [35] to measure SA. Of these methods, EEG has the most predictive power, while respiratory measures were found to have a negative correlation with SA [36].

The most commonly used physiological technique was based on eye tracking. This included signals as blink rates, pupil dilation, but also behavioral characteristics such as fixation rates, dwell times, and saccade frequency to measure SA [36].

However, physiological methods are noisy, show small correlations with SA, and only provide an overall impression of SA rather than per-object SA. The most promising physiological modality was eye gaze, with eye tracking based features forming the best performing predictors of SA. For a full treatment of this topic we refer the reader to Zhang et al. [36].

## B  Situational Awareness Data Collection

We use DReyeVR [14] as the VR-driving simulator. DReyeVR extends the Carla [37] simulator to add virtual reality integration, a first-person maneuverable ego-vehicle, eye tracking support, and several immersion enhancements such as mirrors and sounds. Our physical setup includes a HTC Vive Pro Eye as the head-mounted VR device, which has built-in eye tracking, and an available eye tracking SDK. For our driving hardware we use a Logitech G29 wheel and pedals kit. For driving routes, we use custom routes from several virutal towns shipped with CARLA. Furthermore, we control the traffic in the simulation such that only a single vehicle or two-wheeler enters the FoV of the driver from a single direction at an intersection. If multiple objects enter the driver's FoV from the same direction at the same time, even if the user presses the corresponding directional buttons multiple times, we use manual post-hoc annotation to resolve ambiguities for button press assignment to objects.

### B.1  Instructions provided to participants:

The following prompt was read to participants before they underwent the first trial route. *"Drive safely while following signs to the goal destination. Your main objective is to arrive at the destination as quickly as possible while driving safely. While doing so, you will also perform a secondary task by pushing buttons to indicate which vehicles, pedestrians or two-wheelers (collectively, traffic objects) you have perceived in the environment around you. Anytime you see a new vehicle please press one of the four arrow key on the left side of your steering corresponding to the direction in which they first appeared in your field of view. Similarly, for pedestrians and two-wheelers use the 4 buttons on the right. For each new traffic object you should only press the button once."*

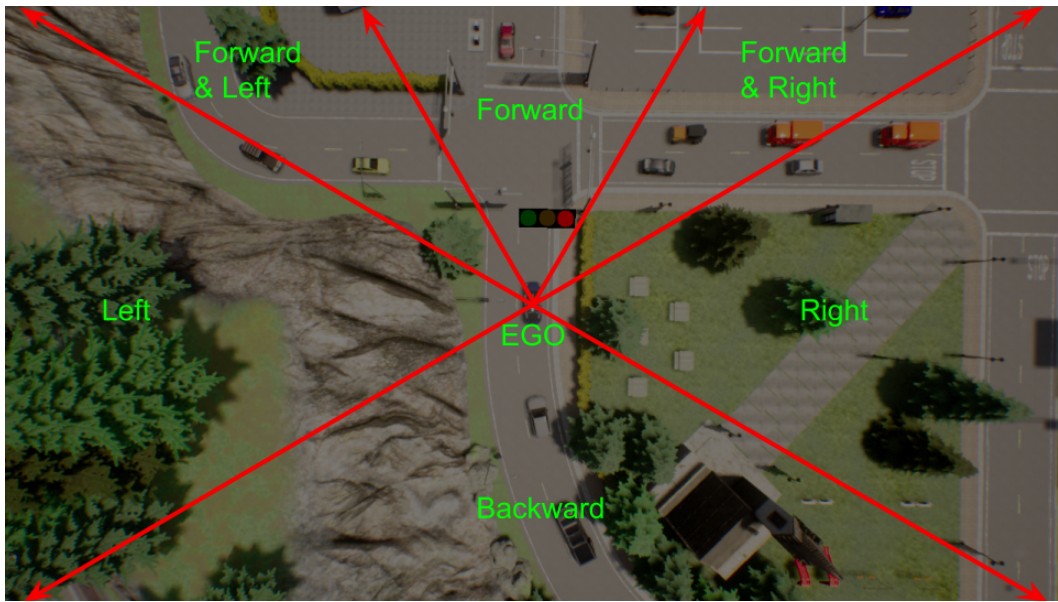

Figure 6: Sectors corresponding to the directions of the button presses. The objects in each sector are target objects for button presses corresponding to the direction of the sector.

## B.2 SA label inference from button presses

Our SA protocol as described in Sec 3 of the paper, allows users to indicate their awareness of objects in the scene using directional button presses. The direction of the button corresponds to the direction of the object. Additionally, there are two sets of directional buttons for the users to choose from. One set corresponds to vehicles and the other set corresponds to pedestrians (+ two-wheelers). For example, when a user first becomes aware of a pedestrian on their left, they would press the left directional button from the button set corresponding to pedestrians.

Our protocol provides us with button clicks, to convert these into awareness labels for object we need to associate button clicks with objects in the scene. We rely on the direction and the set of the button press to associate button presses with objects. We divide the entire scene into 4 sectors corresponding to the 4 directional buttons (Fig. 6). The top sector corresponds to the area between +30 and -30 degree from the ego vehicle. The left sector corresponds to the area between -60 and -120 degree, the right sector corresponds to the area between +60 and +120 degree. The back sector lies between -120 and +120 degrees. The sector between +30 and +60 is considered both forward and right, similarly the sector between -30 and -60 is considered both forward and left.

We keep a track of all the objects that enter each sector, and associate objects with the button clicks pertaining to each sector. The object in each sector, which has not been associated with any button clicks can be associated with a new button click. Objects are considered aware once they are associated with a button click, however once they re-enter of the field-of-view of the driver after leaving it for a certain amount of time, they are again considered unaware and can be associated with button clicks again.

We control the traffic to ensure that there are only a single object of each type (vehicle, pedestrian) in each sector. However, to add randomness we also add a very small number of randomly spawned objects in the scene. Due to this, in certain situations participants' button press inputs can be ambiguous relative to the traffic scene. One common scenario involved multiple potential target objects, in one sector. Additionally, there could also be human errors while pressing buttons, i.e incorrect button type, incorrect direction, or unintentional repeat button presses.To address these ambiguities, we developed a systematic approach to manually evaluate button press instances where the corresponding object was not immediately clear. We examined frames both before and after the button

press, as well as the participant's gaze history, to identify the most likely object associated with the button press.

Currently, our labeling method cannot automatically disambiguate between button presses for candidate objects of the same type that appear at the same time from the same direction. We deal with this issue using human intervention for label correction, in which we asked human observers to review the dataset at points of ambiguity and correct these ambiguous labels, making use of the object masks, RGB, and gaze history. Importantly, these human observers are not solving the same problem as our model later. Instead, they are able to use contextual cues such as preceding button clicks (and assignments), future button clicks (which can make it easier to jointly reason) and future gaze. In contrast to our model, they did not perform awareness inference for every object in the scene, rather they resolved ambiguity within a small number of candidate objects to create a corrected dataset. In the future, experimenters may consider adding an additional requirement of participants providing a small verbal object description accompanying button presses (red vehicle, gray shirt pedestrian) as a disambiguation, especially in very cluttered scenes. We did not consider this labeling approach as asking for these audio descriptions has the potential to influence the gaze away from its natural behavior during driving. Identifying and naming colors may require different cognitive processes (and hence gaze) than simply knowing a vehicle exists towards a given direction. For instance, drivers may fixate on new objects more when they observe them in order to gather information required to generate a description. Furthermore, we found that, in our study, these disambiguations were not very necessary since our scenes were not overly cluttered

### B.3   Route & traffic design:

At least one safety critical scenario such as a jaywalking pedestrian was included in each route. We did so to ensure that driver gaze before and during safety critical scenarios was also represented in the dataset. These types of critical scenarios were included:

1. Visible jaywalking pedestrian: A pedestrian visible without occlusions jaywalks into the ego vehicles path.

2. Simultaneous vehicle turning and jaywalking pedestrian: A vehicle turns left or right while entering at an intersection opposite the ego-vehicle. A pedestrian jaywalks behind the turning vehicle.

3. Occluding object jaywalking pedestrian: A pedestrian, visible from afar but occluded as the ego-vehicle nears, jaywalks into the ego vehicles path.

4. Bicycle crossing after turn: Right after the ego-vehicle makes a right turn, a bicyclist crosses the road in front of the ego vehicle

5. Emergency vehicles distracting from pedestrians: Emergency vehicles are parked near a residence. A policeman, partially occluded by a vehicle, jaywalks to the residence.

See the attached video for examples of critical scenarios.

## C   Modeling Driver SA

**Data representation details:** The virtual camera used to generate visual sensor data for our model was fixed to be $1.3$m above and $1.3$m in front of the ego vehicle (measured from the center of the vehicle base). The camera had a $90°$ field of view and produced $800 \times 600$ images.

**Model and training details:** We used a Feature Pyramid Network [25] segmentation model with a MobileNetV2 [26] backbone (pre-trained on ImageNet). The backbone was chosen for its low number of parameters $(2M)$ and runtime efficiency. Our training procedure used the Adam optimizer with a starting learning rate of $10^{-4}$. The learning rate was scheduled to drop every 5 epochs by a factor of 5.

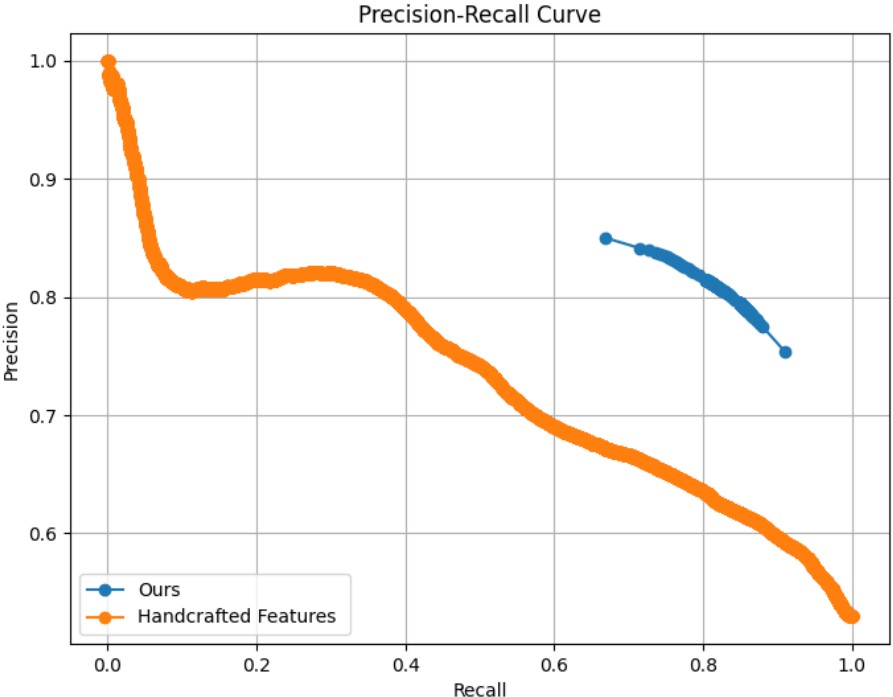

Figure 7: The Precision-Recall curve for our method and the handcrafted-baseline [6]. For our method, we take the mode of predictions over all pixels pertaining to the object, to get the final prediction for the object. To generate the PR curve, our predictions can be thresholded at two levels. First on the raw pixel-level predictions, and second on the ratio of the predicted aware and unaware pixels for a object. Thus, the first threshold level decides what should be the predicted score of a pixel inorder to classify it as aware or unaware. The second level decides how many pixels should be classified as aware inorder to classify this object as aware. To generate this curve we vary the threshold of the raw-pixel level predictions and the second level threshold is fixed at 1. Due to these two levels of thresholds, our method does not have precision = 1 or recall = 1.

## D   Additional Results

A PR curve corresponding to the results in Table 1 in the main paper is shown in Fig. 7.

