# OpenReview forum: "Modeling Drivers’ Situational Awareness from Eye Gaze for Driving Assistance"
_robot-learning.org/CoRL/2024/Conference — CoRL 2024_

### Official Review · Reviewer_NLkr · 2024-07-21
**Sound contribution in data collection and modeling for driver situational awareness**

**Originality:** 4
**Technical Quality:** 3
**Clarity Of Presentation:** 3
**Potential Impact:** 3
**Recommendation:** 3
**Confidence:** 3

**Review:**

Strengths:
1. The paper addresses an important gap in situational awareness modeling, namely availability of labeled datasets linking eye gaze to objects in the scene in a temporally-correlated way.  Current approaches, e.g. SAGAT, label objects sparsely or are prone to inattentional blindness to objects or other non-realistic artifacts (e.g. pausing a simulation for questions).  To cover this gap, this paper proposes a method of collecting online driving data and gaze in a CARLA VR simulator from participants, and their real-time labels associated with specific objects in the scene, labeling in a continuous manner with small burden on the user.  The use of button presses and otherwise crafting the simulation to ensure semantic unambiguity seems a reasonable approach.

2. The paper introduces a learning approach that classifies labels according to their awareness.  The choice of segmenting the image and use of a gaze history map seems to be a valid and practical first step toward solving the problem.  With both dense gaze and dense labels, and ground truth objects, the learning problem is not particularly challenging, and the authors rightly prioritize runtime aspects (i.e. MobileNetV2), without over-complicating their approach.

Weaknesses:
1. As the paper focuses on the practical aspects of the learning approach and data collection, it is still not clear how well the system may work when deployed in practice.  For instance, the object segmentation will carry its own errors and domain mismatch from the CARLA-based system.  Does the system need to be re-trained with real-world data using a similar paradigm, or is it possible that domain transfer directly from the CARLA-based training dataset to the real-world is possible?

2. Some significant portions of the paper are unclear.  For the data collection, it is unclear how well the approach scales, both to scenes involving arbitrary numbers of traffic objects.  From the descriptions in the appendix, the authors claim the need for a "systematic approach" to repair mistakes in the user's input.  What is this approach, and how many corrections were needed to repair the dataset, and does this limit scalability to realistic driving datasets?  Additionally, on the description of the model, some of the modeling choices are again not clear.  For instance, the choice of gaze and windowing approach are not explained in detail, as are the assumptions behind these choices.  Does treating gaze as points with a limited window limit utility, e.g. ignore the effects of working memory, and attention due to peripheral vision?  Does a fixed window size sufficiently account for working memory?  There are also a number of minor typos and grammar errors in the paper that should be fixed.  For instance, "intentional" > "inattentional", "1906s" > "1960s", among others.

**Quality Of The Limitations Section:**

2

**Questions For Rebuttal:**

The authors are encouraged to address the concerns listed in the Weaknesses section, with the most important questions below:
- How can such a gaze-tracking SA alarm be implemented in a practical setup?  What additional ingredients will be required?  Can it operate in systems where gaze and object tracking is more imperfect than in the simulated VR setup, and will a real-world version of the labeling scheme be required?
- Does the windowing approach properly account for working memory of an object or peripheral vision?  And, how does this scale up to arbitrary scenarios with greater traffic clutter?
- Can the labeling approach scale to arbitrary traffic configurations, with arbitrary numbers of objects, and does this, along with errors introduced though user mis-labeling, impact scalability and real-world collection?
- How accurate does the gaze tracking need to be, and what are the practical limitations in both a simulated and non-simulated setup?

**Robotics Focus:**

3

**Summary Of Paper:**

This paper contributes to addressing current gaps in situational awareness modeling for driver assistance in the following respects: (1) introduction of an in-situ labeling protocol for data collection to collect dense target-object labels, and (2) introduction of a learning approach capable of accurately predicting situational awareness over obstacles.

**Summary Of Recommendation:**

Recommend accept, as the paper addresses a significant gap in gaze-based situational awareness, and introduces novel, yet practical approaches whose elements show promise in simulated driving setups.

---

### Official Review · Reviewer_Qeo2 · 2024-07-23
**Strong contributions even with potentially flawed evaluation**

**Originality:** 4
**Technical Quality:** 4
**Clarity Of Presentation:** 4
**Potential Impact:** 3
**Recommendation:** 3
**Confidence:** 4

**Review:**

(++) The application space, ie, that of making driver assistance systems more intelligent through more accurate modelling of human driver situational awareness is well-motivated, interesting, and challenging.



(++) The authors make a number of strong contributions to the community, including piloting a new method of data collection, providing a new dataset, and providing baseline results for a method that leverages this data.



(++) The proposed data collection scheme is intuitive and reasonable. Moreover, it would seem that the annotations it produces are actually predictable using the inputs to the SA prediction system chosen by the authors, which hints that the annotations may actually align with human scene understanding.



(--) The section quantifying the SA prediction system is a little contrived in that the authors have simultaneously proposed a new notion of SA (if only implicitly based on their new labeling paradigm) and a new system for predicting this kind of SA. Therefore, it’s difficult to understand which of these factors is contributing to the prediction results vs. the comparison methods. Is the proposed method superior to [6] even when trained on the dataset used to evaluate [6] or is it somehow only appropriate in the context of this new type of SA? Likewise, if the authors of [6] had been aware of this new definition of SA, would their hand-crafted features have been different? Really the most appropriate comparison would be to use the SA prediction outputs in some downstream system (eg, alert generation as mentioned by the authors in the motivation) and compare some objective metric of that system (eg, user data).

**Post-response update**: Thanks to the authors for answering my questions and agreeing to revise the paper to acknowledge the point about the comparison.

**Quality Of The Limitations Section:**

3

**Questions For Rebuttal:**

(1) Did the authors try their proposed prediction architecture on the dataset used by [6]? Why or why not? If yes, how did it perform relative to the system proposed there?

**Robotics Focus:**

3

**Summary Of Paper:**

The authors consider the overall problem of how to allow computers to build an accurate model of the “situational awareness” of a human driver for, eg, intelligently generating alerts and recommendations. Toward solving this problem, the authors propose a novel data collection/labeling scheme and use it to collect a new dataset. Additionally, the authors use the data to help construct an SA prediction system and quantify the performance of that system.

**Summary Of Recommendation:**

While I have some concerns about the evaluation method here, I do think that the authors have made a strong contribution to the community with a new data collection paradigm and corresponding dataset. The work should be of interest to many in the CoRL community.

---

### Official Review · Reviewer_exse · 2024-07-24
**Generally good -- recommended Weak Accept**

**Originality:** 4
**Technical Quality:** 3
**Clarity Of Presentation:** 5
**Potential Impact:** 3
**Recommendation:** 3
**Confidence:** 5

**Review:**

This paper is of good quality, both in its methodological rigor and its presentation. The proposed method is well-founded on existing literature, and the authors designed their study to address a gap in the field of driving assistance systems. The use of a VR driving simulator to collect detailed and continuous SA data is a strength. The experimental results are robust, with thorough comparisons to baselines. The authors provide a comprehensive introduction that sets the context and clearly outlines the problem. The methodology section is detailed, explaining the interactive labeling protocol and the SA prediction model with precision. The interactive labeling protocol for obtaining continuous and dense SA labels is a novel contribution to the field.

Strengths:
1) The labeling protocol for continuous and dense SA data collection is a major contribution.
2) The SA prediction model is well-designed. The experimental results are comprehensive, with comparisons to multiple baselines.

Weaknesses:
1) The experiments are conducted with a limited number of scenarios and participants. I understand that collecting data from human subject experiments is difficult, but it would be better if this work could collect more data.
2) The labeling protocol may encounter ambiguities in scenarios with multiple objects from the same direction. While the authors mentioned this limitation, I would still suggest that they could elaborate more on this concern since it is important to increase the complexity of the experiments.

**Quality Of The Limitations Section:**

3

**Questions For Rebuttal:**

Questions:
1) As mentioned in the comments, I suggest the authors elaborate more on the labeling ambiguity issue when encountering multiple objects simultaneously and in the same direction. What are the possible solutions?
2) The authors mentioned that “at least one safety critical scenario such as a jaywalking pedestrian was included in each route”. Would participants, human drivers, treat these scenarios differently? Do the authors consider all scenarios together to increase the generalizability? Or should different scenarios be analyzed separately?
3) During data collection, drivers need to press buttons on the wheel. Did drivers look at the wheel while pressing the buttons? If yes, would this process affect their SA? For example, might they miss some objects while looking at the wheel?

**Robotics Focus:**

3

**Summary Of Paper:**

This paper presents a dataset of collected drivers’ situational awareness (SA) in simulation and an approach to improve modeling drivers' SA. The authors propose an interactive labeling protocol that generates continuous and dense object-level SA labels using a VR driving simulator. The constructed dataset includes driving actions, eye gaze data, and SA labels. This work also introduces an SA prediction model formulated as a semantic segmentation problem, enabling the simultaneous processing of all objects in a scene by leveraging global context and local gaze-object relationships. The model demonstrates improved performance over common-sense baselines and previous methods in predicting drivers' awareness of traffic objects. The contributions include the novel SA labeling protocol, the extensive dataset, and the robust SA prediction model.

**Summary Of Recommendation:**

I recommended Weak Accept. This work introduces a novel approach to collect data for drivers' situational awareness. However, while the authors have already addressed some of the limitations in the paper, I would suggest they elaborate more on the possible solutions, even without experiments, to these important concerns.

---

### Decision · Program_Chairs · 2024-09-04

**Decision:**

Accept

**Comment:**

Summary of the paper

This paper addresses the challenge of modeling drivers' situational awareness (SA) to enhance driving assistance systems. The authors propose a novel interactive labeling protocol and a VR driving simulator setup to collect continuous and dense SA data, which includes detailed driving actions and eye gaze information. This innovative dataset is used to train a situational awareness prediction model, framed as a semantic segmentation task, allowing the simultaneous processing of all objects in a scene. The approach shows improved performance over traditional methods and baselines in predicting drivers' awareness of traffic objects, offering a comprehensive framework for intelligent driver assistance.

Strengths
- The paper introduces a novel interactive labeling protocol that collects continuous and dense SA data, filling a significant gap in existing methods that often rely on sparse and intermittent data collection techniques.
- The dataset collected using a VR driving simulator is extensive and includes valuable annotations linking eye gaze to objects, providing a rich resource for further research in situational awareness modeling.
- The proposed SA prediction model is well-designed, utilizing semantic segmentation to leverage both global scene context and local gaze-object interactions, resulting in superior performance compared to existing methods.
- The research addresses a critical gap in driver assistance systems, with potential applications in improving safety and reducing alert fatigue by accurately modeling drivers' situational awareness.
- The study is methodologically sound, with thorough experimental validation and comparisons to various baselines, demonstrating the robustness and efficacy of the proposed model.

Weaknesses
- The reliance on a VR driving simulator may not fully replicate real-world driving conditions, and the model's performance in real-world environments remains untested. The paper would benefit from exploring the feasibility of domain transfer from VR simulations to real-world applications.
- The interactive labeling protocol may encounter challenges with scenarios involving multiple objects from the same direction, potentially leading to labeling ambiguities. Further elaboration on how these ambiguities are resolved and the impact on scalability would strengthen the study.
- The paper lacks clarity on how gaze-tracking accuracy and the proposed SA model would perform in practical, non-simulated setups where tracking might be less precise.
- Some modeling choices, such as the windowing approach and gaze representation, are not clearly justified. Further explanation of these assumptions and their impact on the model's effectiveness would enhance the paper's clarity and comprehensiveness.
- There are minor typographical and grammatical errors in the paper that should be corrected to improve clarity and presentation quality.

Summary of the rebuttal phase

The authors have presented revisions in response to comments from the reviewers, and improvements in the quality of the paper have been acknowledged by the reviewers. Through this rebuttal process, it is believed that their contribution to the CoRL community has become more evident.